# Limitations of Measure-First Protocols in Quantum Machine Learning

**Casper Gyurik** [1 2 3]   **Riccardo Molteni** [1 2]   **Vedran Dunjko** [1 2]

## Abstract

In recent times, there have been major developments in two distinct yet connected domains of quantum information. On the one hand, substantial progress has been made in so-called randomized measurement protocols. Here, a number of properties of unknown quantum states can be deduced from surprisingly few measurement outcomes, using schemes such as classical shadows. On the other hand, significant progress has been made in quantum machine learning. For example, exponential advantages have been proven when the data consists of quantum states and quantum algorithms can coherently measure multiple copies of input states. In this work, we aim to understand the implications and limitations of combining randomized measurement protocols with quantum machine learning, although the implications are broader. Specifically, we investigate quantum machine learning algorithms that, when dealing with quantum data, can either process it entirely using quantum methods or measure the input data through a fixed measurement scheme and utilize the resulting classical information. We prove limitations for the general class of quantum machine learning algorithms that use fixed measurement schemes on the input quantum states. Our results have several implications. From the perspective of randomized measurement procedures, we show limitations of measure-first protocols in the average case, improving on the state-of-the-art which only focuses on worst-case scenarios. Additionally, previous lower bounds were only known for physically unrealizable states. We improve upon this by employing quantum pseu-

dorandom functions to prove that a learning separation also exists when dealing with physically realizable states, which may be encountered in experiments. From a machine learning perspective, our results are crucial for defining a physically meaningful task that shows fully quantum machine learning processing is not only more efficient but also necessary for solving certain problems. The tasks at hand are also realistic, as the algorithms and proven separations hold when working with efficiently preparable states and remain robust in the presence of measurement and preparation errors.

## 1. Introduction

A central question in quantum machine learning revolves around understanding the various types of advantages one can achieve by exploiting quantum effects. Some of the most interesting scenarios arise when the dataset itself comprises quantum states, which can then be processed fully coherently, or through elaborate measurement strategies. In this context, exponential advantages have been identified when coherent measurements of multiple copies of a given quantum state are allowed (Chen et al., 2022b; Huang et al., 2022; 2020; King et al., 2025; Chen et al., 2024a;b). In a parallel related line, there have been significant breakthroughs in extracting useful classical information from quantum states using the versatile toolkit of randomized measurements (Elben et al., 2023). This toolkit includes the groundbreaking concept of classical shadows (Huang et al., 2020; 2022), which can extract an efficient classical description of quantum states that allows one to compute various physical properties. These two distinct research lines cast doubt on the advantages of quantum machine learning protocols that process quantum data directly compared to those that use a fixed measurement procedure, which can also allow for a coherent manipulation of quantum states before the fixed measurement, in order to extract valuable classical data[1]. *In particular we address the question: is it possible*

[1]applied Quantum algorithms, Leiden University, 2333 CA Leiden, Netherlands [2]LIACS, Universiteit Leiden, 2333 CA Leiden, Netherlands [3]Pasqal SaS, 7 rue Léonard de Vinci, 91300 Massy, France. Correspondence to: Casper Gyurik <casper.gyurik@pasqal.com>, Riccardo Molteni <r.molteni@liacs.leidenuniv.nl>, Vedran Dunjko <v.dunjko@liacs.leidenuniv.nl>.

*Proceedings of the 42$^{nd}$ International Conference on Machine Learning*, Vancouver, Canada. PMLR 267, 2025. Copyright 2025 by the author(s).

---

[1]Technically, we allow that the processing of the classical data is also done on a quantum computer. What is critical for the "measure first" stage is that the data is measured out first, before the possible quantum processing.

*that a "measure-first" protocol can be universally used as a substitute for any "fully-quantum" protocol in the general context of machine learning?*

We formally define what we mean by a "measure-first" or a "fully-quantum" protocol in Section 2 (see Definition 2.3 and Definition 2.5), and we provide an overview in Figure 1. Importantly, we focus on physically relevant scenarios where the quantum states in the dataset are efficiently preparable, rather than arbitrary unphysical states that do not allow a polynomial description in terms of quantum gates.

Our main conceptual contribution is to resolve the above question negatively by presenting a concrete machine learning scenario that clearly exhibits the limitations of any measure-first protocol. In a machine learning setup, we construct a task that requires reproducing an undisclosed measurement from the data. This task shows an exponential difference in the number of data needed between fully-quantum protocols and measure-first protocols.

Outside of machine learning, certain limitations of measure-first protocols are implied by other works in various tasks such as distributed sampling (Montanaro, 2019) and relational problems (Aaronson et al., 2024). However, these results primarily address separations in worst-case scenarios, which contrasts with the typical focus in machine learning settings where average-case performance is usually sufficient. In this paper, we advance these findings by identifying the limitations of measure-first protocols that need only achieve average-case correctness for a set of efficiently preparable quantum states, i.e. physical states. On a technical level, establishing limitations for measure-first protocols in a general quantum machine learning setting necessitates novel techniques beyond those utilized in previous studies. In particular, limitations for the task of predicting properties of quantum states were also recently derived for worst-case settings (Grier et al., 2024) exploiting lower bounds for classical one-way communication complexity. However, this approach inherently limits the focus to states that are not efficiently preparable. This limitation arises because, if states were efficiently preparable, there would be a concise classical description of their preparation procedure, enabling the transmission of valid classical messages to the recipient. Our main technical contribution lies in exploiting that, from the more general machine learning standpoint, it makes sense to constrain the computational power of the learning protocol. This allows us to additionally impose the requirement that states must be efficiently preparable and still achieve a separation by utilizing a novel construction that combines results on one-way communication complexity and pseudorandom quantum states (Brakerski & Shmueli, 2019). The task we consider is also experimentally robust, in the sense that we allow for errors in the preparation of the input quantum states and on the measurement outcomes which label them.

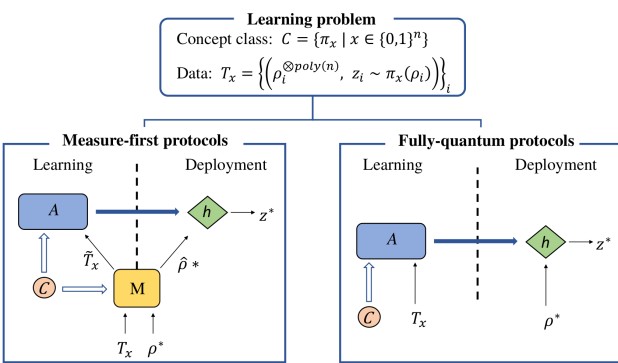

*Figure 1.* An overview of the quantum machine learning protocols explored in our paper. In both protocols, the quantum algorithm $A$ is tasked with processing training data $T_x$ to output a classical description of a function $h$ that correctly produces samples $z^*$ for new input states $\rho^*$ (i.e., $h$ should implement the quantum measurement). The difference between the two protocols is that, in the *measure-first* protocol, both the input quantum states and those in the training set $T_x$ are subjected to a randomized measurement strategy $M$. Consequently, these quantum states $\rho^*$ are transformed into a classical representation $\hat{\rho}^*$. Importantly, this strategy is allowed to depend only on the concept class $\mathcal{C}$ (which captures the general learning problem), not on the specific target concept $\pi_x \in \mathcal{C}$. In other words, the same randomized measurement strategy $M$ is used to preprocess the quantum data for all concepts $\pi_x \in \mathcal{C}$.

## 1.1. High-level overview of learning setting and main result

The machine learning problem concerns learning an unknown measurement acting on a set of input quantum states. Specifically, the data that the learning protocol gets consists of pairs of copies of $n$-qubit quantum states $\rho$ drawn from some distribution $\mathcal{D}$ together with a corresponding label $z \in \{0,1\}^{2n+1}$. The first $n$ bits of $z$ encode a $2^n$-outcome POVM measurement $\Lambda_x = \{E_j^x \mid j \in \{0,1\}^n\}$ from some set $\Lambda = \{\Lambda_x \mid x \in \{0,1\}^n\}$. The remaining $n+1$ bits are determined by the outcome of $\Lambda_x$ on the quantum state $\rho$. The goal of the machine learning protocol is to "learn" how to reproduce the measurement $\Lambda_x$. More precisely, the trained learning protocol has to receive as input an unseen quantum state $\rho'$ and output a sample $z'$ in agreement with the probability distribution $\pi_x(\rho)$, where $\pi_x(\rho)$ denotes the distribution of the measurement outcomes of $\Lambda_x$.

This paper explores whether solving learning problems such as the above requires a quantum computer capable of adaptive measurements on the training data, or if a fixed measurement strategy that produces classical representations of the quantum states is sufficient. Specifically, we consider so-called "*measure-first protocols*" that are forced to measure the input states and use the obtained classical description to train the machine learning model to produce samples

from the target distribution. Importantly, in a measure-first protocol the measurements are not allowed to depend on the training data, but are unconstrained otherwise. In particular, the measure-first protocol can depend on the *set of measurements* that is to be reproduced from the data, but not on the specific *individual measurement* hidden within some given training data. On the other hand, we consider so-called "*fully-quantum protocols*" that can coherently process the quantum states and adjust the measurements based on the training data. Our main result is the existence of a learning problem where the quantum data is efficiently generatable on a quantum computer for which a "measure-first protocol" requires an exponential amount of data to be able to reproduce measurement outcomes on new input states, whereas a fully-quantum protocol only requires a polynomial amount of data. Additionally, we require that the protocols must be efficient in the sense that they run in time polynomial in $n$. Using the notion of learning we presented above, we now give an informal description of the main result of this paper.

**Theorem 1.1.** (Informal) *There exist a concept class $\mathcal{C}$, where each concept is defined by a measurement and a distribution $\mathcal{D}$ over efficiently preparable quantum states such that no "measure-first" protocol can learn to reproduce $\mathcal{C}$ correctly on average with a polynomial amount of training data. On the other hand, there exists a "fully-quantum" protocol which can learn $\mathcal{C}$ efficienctly with respect to both sample and time complexity .*

The specific learning problem which exhibits a learning separation is carefully outlined in Section 3 and the precise proofs of Theorem 1.1 can be found in the Appendix.

## 1.2. Related work

In this section we discuss related works and highlight their relationships to our learning setting. Firstly, in (Huang et al., 2020), the authors introduced a randomized measurement technique tailored to extract a classical description of a quantum state $\rho$. This description enables the computation of the expectation values of any set of observables $\{O_k\}_{k=1}^M$ – provided they have a low "shadow norm", such as when the observables are local – up to a precision of $\epsilon$. Notably, they showed that a number of copies of $\rho$, scaling logarithmically with the number of observables $M$ and inverse-polynomially in the precision $\epsilon$, suffices for this task. As we want to learn the distribution associated of a $2^n$-outcome POVM with an $\epsilon$ error in total variation, directly applying these techniques to estimate each of the $2^n$ outcomes would require an exponential precision for each. In their upper bound estimates, this results in the requirement of an exponential number of samples.

The concept of shadow tomography, introduced in (Aaronson, 2018), revolves around the problem of computing the expectation values of any set of $M$ two-outcome measure-

ments on an $n$-qubit state $\rho$ up to precision $\epsilon$. It has been shown that this can be done using a number of copies of $\rho$ that scale polylogarithmically in $M$, linearly in $n$, and inverse-polynomially in $\epsilon$ (Aaronson, 2018). In contrast to the methods in (Huang et al., 2020), the approach in (Aaronson, 2018) requires coherent measurements on multiple copies of $\rho$. Additionally, as demonstrated in (Huang et al., 2022; Chen et al., 2022a) for specific tasks, the capacity to coherently measure multiple copies of quantum states provides an exponential advantage in sample complexity over sequential measurements. Considering our framework, where measurements can act coherently on multiple copies of each input state, one might question whether such strategies enable a measure-first protocol to solve our learning task. However, it is important to note that coherent measurements do not improve the scaling with respect to the precision $\epsilon$, leaving room for the possibility of a separation since the learning task studied in this paper would require at least exponential precision to be solved.

In (Gong & Aaronson, 2023), the authors extended the concept of shadow tomography to the scenario of learning a $K$-outcome POVMs (for $K \geq 2$) selected from a set of $M$ unknown quantum measurements. In contrast to the binary outcome case, the goal now is to approximate an unknown distribution up to precision $\epsilon$ in total variation distance, rather than focusing on expectation values. Their procedure requires a number of copies of the quantum state scaling linearly with $K$ and $n$, polylogarithmically with $M$, and inverse-polynomially with the precision $\epsilon$. Moreover, they establish the optimality of this scaling with respect to the dependence on the number of outcomes $K$. This result prompts the question of whether our separation between measure-first and fully-quantum protocols can be directly inferred from it. However, in establishing their lower bound, it is important to note that no assumptions were made regarding the complexity of the unknown quantum state. The crux of our study lies in demonstrating that measure-first protocols fall short of reproducing the unknown measurement, even on quantum states that are efficiently preparable. Moreover, it is important to highlight that while their shadow tomography procedures can be employed to construct a measure-first protocol by "shadowfying" input states to approximate the expected values of each measurement outcome, this is not strictly necessary for our task. Specifically, understanding the probability of each outcome allows for the creation of an "evaluator" that can compute the correct probability for every outcome. However, to resolve our learning problem, a "generator" (i.e., an algorithm generating samples with the correct probabilities) already suffices, and it does not necessarily require computing the output probabilities (Sweke et al., 2021).

Next, in (Grier et al., 2024) the authors explore lower bounds on the number of copies of a quantum state required for solv-

ing what they term the "classical shadow" task. This task involves estimating the expected values of observables using a measure-first approach. While they establish lower bounds on the number of copies of a quantum state required for this task, shedding light on the inherent limitations of measure-first protocols, it is crucial to note two significant bottlenecks of their approach when viewed from our more general machine learning perspective. Firstly, their analysis only focuses on worst-case scenarios, whereas in our machine learning scenarios we are concerned with average-case performance. Secondly, their investigation centers on protocols required to operate effectively even for unphysical quantum states that are not efficiently preparable.

The authors of (Cheng et al., 2016) provide upper bounds for the dual problem of shadow tomography, or more specifically the problem of learning a measurement. In particular, they studied the task of learning an unknown two-outcome POVM denoted $\{E, I - E\}$, from data of the form $\{(\rho_i, \text{Tr}(E\rho_i))\}_{i=1}^m$. They showed that to approximate the unknown $E$ up to a precision of $\epsilon$ on new $n$-qubit input states, $\mathcal{O}(2^n/\epsilon^2)$ training samples are sufficient. Nonetheless, the number of required samples scales exponentially with the number of qubits.

Finally, in (Jerbi et al., 2023) the authors study quantum process learning, where the task is to learn a unitary $U$, from data of the form $\{|\psi_i\rangle, U|\psi_i\rangle\}$. In particular, they study the limitations of what they call incoherent learning, where the learner is constrained to first measure multiple copies of the data $U|\psi_i\rangle$. While they therefore also study the problem of extracting classical information from quantum data and utilizing it in the learning process, the setting in their work differs from ours. Namely, the quantum states in our scenario are labeled by a sample obtained from the unknown measurement process, whereas in (Jerbi et al., 2023) the labels are the input quantum states when evolved under some unknown target unitary.

## 2. Background

In this section, we introduce the notations and definitions needed to frame our main result, which involves a learning problem that demonstrate the separation between any measure-first protocol and a fully quantum protocol.

The learning problem we study is the learning of a measurement. In particular, it involves generating samples from a distribution induced by measuring an (unknown) POVM measurement $\Lambda_x$ on $n$-qubit quantum states. The (unknown) target measurement $\Lambda_x$ is drawn from a set of POVMs $\{\Lambda_x \mid x \in \{0,1\}^n\}$, and each measurement $\Lambda_x$ is a computational basis measurement preceded by an $n$-qubit unitary $U_x$, i.e.,

$$\Lambda_x = \left\{ E_j^x \mid j = 1, \ldots, 2^n \right\}, \text{ where } E_j^x = U_x |j\rangle \langle j| U_x^\dagger.$$

During training the learner is given a set of examples $T_x$, where each example consists of a polynomial number of copies of a phase state $|\psi_f\rangle$ together with a sample from the associated POVM-induced distribution. This is a special case of labeled quantum data, which was introduced in (Aïmeur et al., 2006), where we are additionally allowed to have access to polynomially many copies of the input quantum state. We formalize our learning problem by generalizing the standard PAC learning framework (Valiant, 1984). In our generalization, a concept corresponds to a quantum randomized function, i.e., a function that on each quantum input state outputs a sample from a random variable (which in our case corresponds to the outcomes of a POVM on the input quantum state). Before we define the concept class studied throughout this paper, we first setup some auxilliary definitions.

**Definition 2.1** (Auxiliary definitions/notation).

- Let $N = 2^n$, or equivalently $n = \log_2 N$.

- We identify a function $f : \{0,1\}^n \to \{0,1\}$ with its truth table $f \in \{0,1\}^N$, and we denote its corresponding *phase state* with

$$|\psi_f\rangle = \frac{1}{\sqrt{N}} \sum_{i=1}^N (-1)^{f_i} |i\rangle. \tag{1}$$

- Let $\mathcal{S}_{\text{phase}} = \{|\psi_f\rangle^{\otimes \ell} \mid f \in \{0,1\}^N\}$ denote the set of $\ell = \text{poly}(n)$ copies of $n$-qubit phase states.

- We write $x \sim \pi$ to denote that $x$ was drawn according to a distribution $\pi$.

- We write $\mathcal{U}(\mathcal{X})$ for the uniform distribution over a set $\mathcal{X}$.

- We write $\Delta(\mathcal{X})$ for the set of all distributions over a set $\mathcal{X}$.

- For $p, q \in \mathbb{R}_{\geq 0}^{2^n}$ probability distributions over $\{0,1\}^n$ we define

$$||p - q||_{TV} = \frac{1}{2} \sum_{x \in \{0,1\}^n} |p_x - q_x|$$

to be the *total variation distance*.

**Definition 2.2** (Concept class). We define our concept class as $\mathcal{C} = \{\pi_x \mid x \in \{0,1\}^n\}$, where

$$\begin{aligned} \pi_x : \mathcal{S}_{\text{phase}} &\to \Delta(\{0,1\}^{2n+1}) \\ |\psi_f\rangle^{\otimes \ell} &\mapsto \pi_x(f) \end{aligned} \tag{2}$$

where $\pi_x(f)$ is a distribution over samples $(x, y, b) \in \{0,1\}^{2n+1}$, where $(y, b) \in \{0,1\}^{n+1}$ are variables sampled from a problem-specific distribution characterizing the specific learning task.

In particular, $\pi_x$ is a randomized function which takes as input a polynomial number of copies of a phase state and outputs a sample $z$ consisting of $x \in \{0,1\}^n$ together with some $(y,b) \in \{0,1\}^{n+1}$ sampled from some problem-specific probability distribution. As we will see in the next section, a learning separation for this kind of learning problem is achieved by considering $(y,b)$ sampled uniformly from the set of variables which satisfy the Hidden Matching relation defined by the input functions $f$ and the bitstring $x \in \{0,1\}^n$.

A learner then is given several evaluations of the randomized function $\pi_x$ in the form of training data and its objective is to implement a randomized function $\widetilde{\pi}_x$ that closely approximates $\pi_x$ on most input states.

In this paper, we compare two categories of machine learning systems that can tackle problems of this type. First, we introduce what we call a "*fully-quantum protocol*".

**Definition 2.3** (Fully-quantum protocol). A *fully-quantum protocol* for the concept class $\mathcal{C}$ in Definition 3.1 is a polynomial-time quantum algorithm $A$ that takes as input training data of the form

$$T_x = \left\{ \left( |\psi_{f^{(i)}}\rangle^{\otimes \ell}, (x,y,b) \right) \mid (x,y,b) \sim \pi_x(f^{(i)}), \right.$$
$$\left. \text{and } f^{(i)} \sim \mathcal{U}\left(\{0,1\}^N\right) \right\}_{i=1}^{\text{poly}(n)},$$
$$(3)$$

and outputs a *classical description of a polynomial-time quantum algorithm* that on input $|\psi_f\rangle^{\otimes \ell} \in \mathcal{S}_{\text{phase}}$ generates a sample from a distribution $\widetilde{\pi}_x(f) \in \Delta(\{0,1\}^{2n+1})$.

We emphasize that for a "fully quantum" protocol, the learning algorithm must produce a *classical description* of the quantum algorithm generating samples from $\widetilde{\pi}_x$. Consequently, we do not store any quantum states from the training data in quantum memory, which would be more general but not studied in this paper. Ultimately, the goal of the protocol is to implement a randomized function $\widetilde{\pi}_x$ that closely approximates the actual data-generating randomized function $\pi_x$ for most of the input quantum states.

**Definition 2.4** (($\epsilon, \delta, p_{\text{succ}}$)-fully-quantum learnable). We say that $\mathcal{C}$ is ($\epsilon, \delta, p_{\text{succ}}$)-*fully-quantum learnable* if there exists a fully-quantum protocol $A$ such that for every $\pi_x \in \mathcal{C}$, with probability at least $p_{\text{succ}}$ we have

$$\Pr_{f \sim \mathcal{U}(\{0,1\}^N)} \left( ||\widetilde{\pi}_x(f) - \pi_x(f)||_{TV} \leq \epsilon \right) \geq 1 - \delta, \quad (4)$$

where $\widetilde{\pi}_x(f) \in \Delta(\{0,1\}^{2n+1})$ denotes the distribution that the polynomial-time quantum algorithm obtained from the learning algorithm $A$ generates samples from on input $|\psi_f\rangle^{\otimes \ell} \in \mathcal{S}_{\text{phase}}$.

Next, we introduce a "*measure-first protocol*" which consists of two components: (i) a randomized measurement

strategy $M$, and (ii) a learning algorithm $A$. The main difference between a measure-first protocol and a fully-quantum protocol is that the former involves a randomized measurement procedure that first measures the quantum states before putting it into a learning algorithm. Importantly, the measure-first protocol is allowed to perform arbitrary coherent measurements on all input quantum states (i.e., the polynomially-many copies of the phase states). The only constraint is that the *measurement strategy cannot depend on the specific target concept* of the learning problem, although it is allowed to depend on the concept class (i.e., the set of all possible target concepts). In short, a randomized measurement strategy is a polynomial-time algorithms that maps a polynomial number of copies of a phase state $|\psi_f\rangle$ to some classical description $\widehat{\psi}_f \in \{0,1\}^m$ for some $m = \text{poly}(n)$. These classical descriptions $\widehat{\psi}_f$ are then used as input for the learning algorithm, that is tasked with implementing a randomized function close to $\pi_x$.

**Definition 2.5** (Measure-first protocol). A *measure-first protocol* is a tuple $(M, A)$ where

- $M$ is a measurement strategy that in time $\mathcal{O}(\text{poly}(n))$ maps $|\psi_f\rangle^{\otimes \ell} \in \mathcal{S}_{\text{phase}}$[2] to some $\widehat{\psi}_f \in \{0,1\}^m$, where $m = \text{poly}(n)$.

- $A$ is a polynomial-time quantum algorithm that takes input of the form

$$T_x^M = \left\{ \left( \widehat{\psi}_{f^{(i)}}, (x,y,b) \right) \mid (x,y,b) \sim \pi_x(f^{(i)}) \right.$$
$$\left. \text{and } f^{(i)} \sim \mathcal{U}\left(\{0,1\}^N\right) \right\}_{i=1}^{\text{poly}(n)},$$
$$(5)$$

and outputs a *description of a polynomial-time quantum algorithm* that on input $\widehat{\psi}_f = M(|\psi_f\rangle^{\otimes \ell})$ generates a sample from a distribution $\widetilde{\pi}_x(f) \in \Delta(\{0,1\}^{2n+1})$.

Note that the distinction between measure-first and fully-quantum protocols lies in Eq. 5, where the data is measured to give $\widehat{\psi}_f$ instead of remaining quantum states $|\psi_f\rangle$. Nonetheless, the measurements employed by the measurement strategy $M$ are completely arbitrary and entirely unrestricted. We emphasize, however, that the sole constraint on the measurement strategy $M$ is that it must not rely on the specific concepts it will be applied to later, although it can depend on the concept class it is intended for. Recall that the objective of the protocol is to implement a randomized function $\widetilde{\pi}_x$ that closely approximates the actual data-generating randomized function $\pi_x$ on most inputs.

---

[2] For completeness, we note here that our results still apply if $M$ is allowed to coherently measure multiple input data for different $f$ and then return a polynomial-size classical representation.

**Definition 2.6** $((\epsilon, \delta, p_{\text{succ}})$-measure-first learnable)**.** We say that $\mathcal{C}$ is $(\epsilon, \delta, p_{\text{succ}})$-*measure-first learnable* if there exists a measure-first protocol $(M, A)$ such that for every $\pi_x \in \mathcal{C}$, with probability at least $p_{\text{succ}}$ we have

$$\Pr_{f \sim_U \{0,1\}^N} \left( ||\widetilde{\pi}_x(f) - \pi_x(f)||_{TV} \le \epsilon \right) \ge 1 - \delta, \quad (6)$$

where $\widetilde{\pi}_x(f) \in \Delta(\{0,1\}^{2n+1})$ denotes the distribution that the polynomial-time quantum algorithm obtained from the learning algorithm $A$ generates samples from on input $\widehat{\psi}_f = M(|\psi_f\rangle^{\otimes \ell})$.

Although we defined the two protocols in an idealized case, we will show in Appendix A.1 that our separation results are robust. In particular, they still hold even in the experimental setting where the input states $|\psi_f\rangle$ are affected by preparation errors and they are just close in trace distance to the ones defined in Eq.( 1). Furthermore, we also allow for measurement errors on the labels $(y, b)$.

# 3. Main result

In this section, we present the key findings of our paper. In Section 3.1, we define the specific learning problem considered and show how the fully-quantum machine learning model can efficiently solve it. In Section 3.2, we present our first main result showing that no measure-first quantum machine learning model can solve our learning problem efficiently. Finally, in Section 3.3, we show that this separation between the models still holds if the quantum states in the data are efficiently preparable.

## 3.1. The learning problem

In this section, we concretely define the learning problem for which we prove a learning separation. Then, we initiate the proof by showing that it is learnable using a fully-quantum protocol.

The specific concept class we developed to prove our main result is a particular instance of the one defined in Def. 2.2, with the condition that the variables $(y, b)$ are sampled from the set $R_x(f)$ defined as follows:

**Definition 3.1** (Concept class)**.** $\mathcal{C} = \{\pi_x \mid x \in \{0,1\}^n\}$, where

$$\pi_x : \mathcal{S}_{\text{phase}} \to \Delta(\{0,1\}^{2n+1})$$
$$|\psi_f\rangle^{\otimes \ell} \mapsto \pi_x(f) \quad (7)$$

where $\pi_x(f)$ is a distribution over samples $(x, y, b)$, where $(y, b) \sim \mathcal{U}(R_f(x))$ and

$$R_f(x) = \{(y, b) \mid y \in \{0,1\}^n, b \in \{0,1\},$$
$$f(y) \oplus f(y \oplus x) = b\}. \quad (8)$$

Importantly, in (Aaronson et al., 2024) the authors showed that for each $x \in \{0,1\}^n$ there exist a POVM measurement $\Lambda_x$ which when applied to a phase state $|\psi_f\rangle$ outputs a pair exactly satisfying the relation $R_f(x)$. With regards to our learning problem, the task of the learning protocols is to learn this measurement.

We now describe how the concept class in Definition 3.1 is fully-quantum learnable.

**Proposition 3.2.** *The concept class in Definition 3.1 is* $(0, 0, 1)$-*fully-quantum learnable.*

The proof of Proposition 3.2 can be found in Appendix A, and we provide a high-level overview of the fully-quantum protocol here. Firstly, the fully-quantum protocol reads out $x$ from one of the samples generated by $\pi_x$ in the training data. Next, a quantum circuit denoted as $U_x$ is constructed following the procedure outlined in (Aaronson et al., 2024). Measuring $U_x |\psi_f\rangle$ in the computational basis results in a sample from the distribution $\mathcal{U}(R_f(x))$. Crucially, it is worth noting that these quantum circuits $U_x$ are of size $\mathcal{O}(\text{poly}(n))$ and can be constructed in time $\mathcal{O}(\text{poly}(n))$. We remark here that the concept class in Definition 3.1 remains fully quantum learnable even when errors are present in the training data, see Appendix A.1.

It might seem that little genuine learning occurs when $x$ can be readily read out from a single example in $T_x$. However, we can introduce various levels of learning by providing only partial information about $x$ within the examples. This partial information should allow the recovery of $x$ from a polynomial number of examples. Several examples illustrating this are discussed in more detail in Appendix A.

## 3.2. Limitations of measure-first protocols with general quantum data

In the last section, we discussed how the concept class in Definition 3.1 is fully-quantum learnable. Conversely, in this section we discuss the first part of our main result which states that this concept class is *not* measure-first learnable.

**Theorem 3.3.** *The concept class in Definition 3.1 is not* $(\epsilon, \delta, p_{\text{succ}})$-*measure-first learnable for* $(1-\epsilon) \cdot (1-\delta) > 7/8$ *and any* $p_{\text{succ}} > 0$.

The full proof of Theorem 3.3 is provided in Appendix B, we present a concise overview of the proof here. At its core, the proof hinges on the notion that the existence of a measure-first protocol for the concept class described in Definition 3.1 implies the existence of an efficient classical one-way communication protocol for the Hidden Matching (HM) problem (Bar-Yossef et al., 2004). Notably, in (Bar-Yossef et al., 2004), it has been shown that the HM problem cannot be solved with a communication cost of $\mathcal{O}(\text{poly}(n))$ bits, even on a 7/8 fraction of possible inputs. In essence, if the concept class in Definition 3.1 was learnable, one of the

two parties can employ the measurement protocol to encode their input for the HM problem, transmit it to the other party, who can then utilize the learning algorithm $A$ to successfully solve the HM problem. Intuitively, the reason behind why measure-first learning fails is that due to (Bar-Yossef et al., 2004) it is not possible to compress a phase state $|\psi_f\rangle$ into a polynomially-sized classical representation $\widehat{\psi}_f$ that contains enough information to allow one to generate samples from the distributions $\pi_x(f)$ for all possible $x$. Importantly, within the machine learning context, it is crucial to also consider the possibility of protocols making errors on a fraction of inputs and the requirement to gather the necessary data. To overcome this limitation, we use Yao's principle to show that the existence of a protocol which succeeds in the machine learning context would still violate the classical hardness of the HM problem. In particular, by applying Yao's principle, we can ensure the existence of a deterministic algorithm that solves the HM problem on a sufficiently large fraction of inputs, provided that there exists a machine learning measure-first protocol capable of solving the learning task defined in Definition 2.6.

Since in our machine learning setting we are concerned with physical states appearing in the real-world, in the next section we show that our findings also apply to settings where states are efficiently preparable, which we achieve by using pseudorandom states (Brakerski & Shmueli, 2019) (see Section 3.3). Finally, it is crucial to note that even when the data in Eq.( 5) is derived from a polynomial number $\ell$ of copies of each input state $|\psi_f\rangle$, measure-first protocols remain incapable of solving the task. This holds true even when considering inefficient ( i.e., superpolynomial-time ) algorithms or an exponential number of training points, as the communication complexity bounds still apply regardless of the time resources utilized by the algorithms (since the communication protocol assumes both parties have unbounded resources).

## 3.3. Learning separation with efficiently preparable quantum data

From a pragmatic perspective, a crucial limitation of the learning problem outlined in the previous section is that preparing a general phase state is intractable (i.e., not realized by polynomial-time processes). In particular, this raises the question of whether separations could persist for states that are prepared by (natural or artificial) polynomial-time processes. To address this limitation, we show that the concept class in Definition 7 remains not measure-first learnable, even when we constrain the input of the random functions $\pi_x$ to phase states of so-called *pseudorandom functions*. Notably, phase states corresponding to appropriately chosen pseudorandom functions can be efficiently prepared. Because of this additional consideration, our separations are notably more general than those in previous works (Aaron-

son et al., 2024; Bar-Yossef et al., 2004). Our definition of pseudorandom functions is as follows.

**Definition 3.4** (Quantum-secure pseudorandom function (QPRF) (Brakerski & Shmueli, 2019)). Let $\mathcal{K} = \{\mathcal{K}_n\}_{n \geq 1}$ be an efficiently samplable key distribution, and let $\mathsf{PRF} = \{\mathsf{PRF}_n\}_{n\geq1}$, $\mathsf{PRF}_n : \mathcal{K}_n \times \{0,1\}^n \to \{0,1\}$ be an efficiently computable function. We say PRF is a quantum-secure pseudorandom function if for every efficient non-uniform quantum algorithm $A$ that can make quantum queries there exists a negligible function $\mathrm{negl}(.)$ such that for every $n \geq 1$:

$$
\left| \mathsf{Pr}_{k \sim \mathcal{U}(\mathcal{K}_n)} \left[ A^{\mathsf{PRF}_n(k)}() = 1 \right] - \right.
$$

$$
\left. \mathsf{Pr}_{f \sim \mathcal{U}(\{0,1\}^n)} \left[ A^f() = 1 \right] \right| \leq \mathrm{negl}(n) \tag{9}
$$

Where the notation $A^f()$ stands for the non-uniform quantum algorithm which can make queries to the function $f$ and can take any quantum state as input. We remark that if every function $\mathsf{PRF}_n$ admits a classical circuit of size $s(n)$ and depth $d(n)$, then one can prepare the corresponding phase states using a quantum circuit of size $\mathcal{O}(s(n))$ and depth $d(n) + 1$ (Brakerski & Shmueli, 2019). Moreover, the existence of such $\mathsf{PRF}_n$ is implied by the existence of quantum secure one-way functions (Zhandry, 2012).

### 3.3.1. FULLY-QUANTUM LEARNABILITY WITH PSEUDORANDOM PHASE STATES

Note that when we constrain the inputs of $\pi_x$ to phase states of pseudorandom functions, we essentially modify the distribution over input states in Eq. 4 and Eq. 6. This new distribution now only has support on phase states that are efficiently preparable. While Proposition 3.2 examines general quantum phase states as input states (which are not typically efficiently preparable), we note that the fully-quantum learnability directly extends the setting where we limit ourselves to efficiently preparable phase states as well. We summarize this observation in the following proposition (whose proof is the same as that of Proposition 3.2).

**Proposition 3.5.** *Let* $\mathcal{S}_{\mathrm{pr}} = \{|\psi_{f^{(k)}}\rangle^{\otimes \ell} \mid f^{(k)}(.) = \mathsf{PRF}_n(k,.), k \in \mathcal{K}\}$, *where* PRF *is a quantum-secure pseudorandom function with keys* $\mathcal{K}$. *The concept class in Definition 3.1 is* $(0, 0, 1)$-*fully-quantum learnable when the distribution over input states is uniform over* $\mathcal{S}_{\mathrm{pr}}$.

### 3.3.2. LIMITATIONS OF MEASURE-FIRST PROTOCOLS WITH PSEUDORANDOM PHASE STATES

In the last section, we discussed how the concept class in Definition 3.1 remains fully-quantum learnable when restricted to phase states of pseudorandom functions. Conversely, in this section we show that this concept class also

remains *not* measure-first learnable when restricted to phase states of pseudorandom functions.

**Theorem 3.6.** *Let $\mathcal{S}_{\mathrm{pr}} = \{|\psi_{f^{(k)}}\rangle^{\otimes \ell} \mid f^{(k)}(.) = \mathrm{PRF}_n(k,.),\ k \in \mathcal{K}\}$, where $\mathrm{PRF}$ is a quantum-secure pseudorandom function with keys $\mathcal{K}$. The concept class in Definition 3.1 is not $(\epsilon, \delta, p_{\mathrm{succ}})$-measure-first learnable for $(1 - \epsilon) \cdot (1 - \delta) \cdot p_{\mathrm{succ}} > c$ for any constant $c > 7/8$ when the distribution over input states is uniform over $\mathcal{S}_{\mathrm{pr}}$.*

The main results of this section directly follows from combining Theorem 3.6 and Proposition 3.5.

**Corollary 3.7** (informal). *If there exist quantum-secure pseudorandom functions, then there exists a quantum supervised learning problem with efficiently generatable quantum data, which cannot be learned by any measure-first protocol according to Def. 2.6 while there exist a fully-quantum protocol which satisfies the learning condition of Def. 2.4*

The proof of Theorem 3.6 is provided in Appendix C, and we first present a concise overview of the proof here. The main idea behind the proof is to illustrate that if the concepts are measure-first learnable when restricted to pseudorandom phase states, then the corresponding measure-first learning protocol can be harnessed to create a non-uniform quantum algorithm that is able to distinguish between truly random functions and pseudorandom functions. More precisely, this "distinguisher" algorithm employs the measure-first learning protocol and evaluates its performance when applied to the phase state corresponding to the function it has been given oracular access to. In the proof of Theorem 3.3, we established an upper bound on the generalization performance of any measure-first protocol for truly random phase states. If, however, the measure-first protocol performs well on pseudorandom phase states, then the outcomes of the "distinguisher" algorithm would differ significantly based on whether it is given oracular access to a truly random or a pseudorandom function, thereby contradicting the pseudorandomness assumption. In other words, if the measure-first protocols were effective on pseudorandom states, there would be a clear difference in the performance of the distinguisher: poor accuracy when dealing with truly random functions and strong accuracy when dealing with pseudorandom functions. However, since this would violate Eq. 9, measure-first protocols have to fail when applied to pseudorandom functions.

## 4. Conclusion

In our study, we explored the constraints and capabilities of learning from quantum data. We established a formal machine learning framework that contrasts two protocols: "fully quantum", which adjusts measurements based on data, and "measure-first" restricted by fixed initial (though arbitrarily powerful) measurements. In particular, we pro-

vided an example of a learning problem efficiently solved by a fully-quantum protocol but beyond the capabilities of measure-first protocols. Moreover, we showed that this persists even when we limit from universal quantum states, which include also those intractable to prepare, to efficiently preparable quantum states. These findings underscore the crucial role of processing unmeasured quantum data in machine learning, presenting a setting where quantum advantages arise. In particular, they imply that certain learning tasks inherently require the "exponential capacity" of quantum states, distinct from classical data. In other words, the number of bits needed to store $n$ qubits, in a way that allows a learner to successfully solve the learning problem, is exponential in $n$. Such a conclusion is analogous to what Montanaro refers as "anti-Holevo" theorems (Montanaro, 2019). While our proof relies on both separations in one-way communication complexity and pseudorandom states, we highlight the potential for more general constructions. Instead of considering states demonstrating a one-way communication complexity separation, any quantum advice state used in quantum advice complexity classes that cannot be classically simulated with polynomial overhead could suffice, leveraging the separation between the classes $\mathrm{FBQP}/\mathrm{qpoly}$ and $\mathrm{FBQP}/\mathrm{poly}$ showed in (Aaronson et al., 2024). In particular, by the result in (Aaronson & Drucker, 2010) any problem that can be efficiently solved with a polynomial-sized quantum advice state can also be solved with an advice state that is the ground state of a local Hamiltonian. This suggests that learning problems where the input quantum data consists of ground states of sufficiently complex local Hamiltonians are promising candidates for demonstrating a separation between measure-first and fully-quantum protocols. Finally, we note that beyond pseudorandom states one could use so-called "computationally indistinguishable" states, which are known to exist assuming the intractability of the graph isomorphism problem (Kawachi et al., 2012), or various other complexity theoretic assumptions (Brakerski et al., 2022).

## Acknowledgements

VD and CG acknowledge the support of the Dutch Research Council (NWO/ OCW), as part of the Quantum Software Consortium programme (project number 024.003.037). This work was supported by the Dutch National Growth Fund (NGF), as part of the Quantum Delta NL programme. This publication is also part of the project Divide & Quantum (with project number 1389.20.241) of the research programme NWA-ORC which is (partly) financed by the Dutch Research Council (NWO).

## Impact Statement

This paper presents work whose goal is to advance the field of Quantum Machine Learning. There are many potential societal consequences of our work, none which we feel must be specifically highlighted here.

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

# A. Proof of Proposition 3.2

**Proposition 3.2.** *The concept class in Definition 3.1 is $(0, 0, 1)$-fully-quantum learnable.*

*Proof.* To prove that the concept class $\mathcal{C}$ in Definition 3.1 is fully-quantum learnable we will provide a fully-quantum protocol $A$ that does so successfully. Suppose we are given training data $T_x$ of the form provided in Eq. 3. Firstly, the fully-quantum protocol $A$ reads out $x$ from one of the examples in $T_x$. Next, it uses the construction of (Aaronson et al., 2024) to construct a circuit $U_x$ of size $\mathcal{O}(n)$ in time $\mathcal{O}(n)$ such that when measuring the state $|\phi_{f,x}\rangle = U_x |\psi_f\rangle$ in the computational basis it produces $(y, b) \in \{0, 1\}^{n+1}$ such that $b = f(y) \oplus f(x \oplus y)$. The operator $U_x$, whose graphical representation is provided in Figure 3 of (Aaronson et al., 2024), is defined as follows. Consider an $x \in \{0, 1\}^n$ with a Hamming weight of $k \geq 1$. The operator $U_x$ acts on $n$ qubits based on the values of the entries of $x$ as follows:

1. A position $i$ is selected such that $x_i = 1$. The corresponding qubit $i$ is then chosen to output the value of $b \in \{0, 1\}$.

2. For each of the remaining positions $j$ where $x_j = 1$ and $j \neq i$, a CNOT gate is applied with qubit $j$ as the control and qubit $i$ as the target. This results in a total of $k - 1$ CNOT gates.

3. Finally, a Hadamard gate is applied to qubit $i$.

Measuring the resulting state $U_x |\psi_f\rangle$ in the computational basis yields an $n$-bit string $j_1 j_2, ..., j_n$ such that the variables $y = j_1, ..., j_{n-1}$ and $b = j_n$[3] are guaranteed to satisfy the relation $b = f(y) \oplus f(x \oplus y)$.

The learning protocol outputs the description of the POVM measurement

$$\Pi_x = \left\{ U_x |j\rangle \langle j| U_x^\dagger \mid j \in \{0, 1\}^n \right\} \tag{10}$$

as by the above measuring $\Pi_x$ on an arbitrary phase state $|\psi_f\rangle$ implements $\pi_x$ with zero error.

While it may appear that little learning is occurring when we can readily extract $x$ from a single example in $T_x$, we can introduce varying degrees of learning by not appending the complete description of $x$ to the examples. Instead, we include only partial information about $x$ that still allow us to recover a full description of $x$ using a polynomial number of examples. For instance, instead of appending $x$ to the examples we can append certain functions $g_i(x)$, where $g_i$ is drawn uniformly random from some set of $\mathcal{G} = \{g_i\}_{i \in I}$. For instance, for $i \in \{0, 1\}^n$ we can consider functions like

$$g_i(x) = (i \cdot x, i) \in \{0, 1\}^{n+1}, \tag{11}$$

where $x \cdot i = \sum_{j=1}^n x_j \cdot i_j \mod 2$. Another example of such a family of functions would be

$$g_i(x) = (\mathrm{DLP}(x)_i, i) \in \{0, 1\}^{n+1}, \tag{12}$$

where $\mathrm{DLP}_i(x)$ denotes the $i$th bit of the discrete logarithm of $x$ in a suitably chosen group. For these functions, one can show that $x$ can be recovered with high probability from a polynomial number of evaluations of $g_i(x)$ for randomly chosen $g_i$ from $\mathcal{G}$. Moreover, functions similar to the $g_i$ in Eq. 12 require a quantum computer to be able to efficiently recover $x$ (Liu et al., 2021).

$\square$

## A.1. Learnability with noisy data

In a realistic setting, quantum states and measurements will always be affected with experimental errors. In this section we study whether the fully-quantum protocol still manages to solve the learning task even in non ideal scenarios.

We specifically address the scenario where the input states are not exactly the phase states $|\psi_f\rangle$ from Eq.( 1 ) but are instead approximate states $|\psi_f\rangle_{\epsilon_{sp}}$ in trace distance, such that:

$$d(|\psi_f\rangle, |\psi_f\rangle_{\epsilon_{sp}}) \leq \epsilon_{sp} \tag{13}$$

---

[3]Without loss of generality, we assume the qubit $i$ in the above construction to be at the positon $i = n$.

where $d(|\psi_f\rangle, |\psi_f\rangle_{\epsilon_{sp}}) = || |\psi_f\rangle \langle\psi_f| - |\psi_f\rangle \langle\psi_f|_{\epsilon_{sp}} ||_1$ is the trace distance between the states $|\psi_f\rangle$ and $|\psi_f\rangle_{\epsilon_{sp}}$.

As mentioned before, we also allow for measurement errors on the training labels. Specifically we can consider the case that the labels $(y, b)$ in the training data follow an approximate distribution $\pi_x^{\epsilon_M}(f)$ which is only close to the ideal one in total variation distance:

$$\forall f \in \{0,1\}^N \qquad ||\pi_x^{\epsilon_M}(f) - \pi_x(f)||_{TV} \leq \epsilon_M \tag{14}$$

Note that only the two variables $(y, b)$ are affected by measurement errors, while the variable $x$, which labels each concept, is not affected by any error.

Even with both of the above sources of experimental error, the learning problem in Def. 3.1 remains fully quantum learnable in the following sense.

**Proposition A.1.** *Assuming a maximum error $\epsilon_{sp}^{max}$ on the input states and a maximum error $\epsilon_M^{max}$ on the corresponding labels, then there exists a fully-quantum protocol $A$ such that for every $\pi_x \in \mathcal{C}$ with probability 1 satisfies:*

$$Pr_{f \sim \mathcal{U}(\{0,1\}^N)} \left( ||\tilde{\pi}_x(f) - \pi_x(f)||_{TV} \leq \epsilon_{sp}^{max} \right) = 1 \tag{15}$$

*where $\tilde{\pi}_x(f) \in \Delta(\{0,1\}^{2n+1})$ denotes the distribution that the polynomial-time quantum algorithm obtained from the learning algorithm $A$ generates samples from on input $|\psi_f\rangle_{\epsilon_{sp}}^{\otimes l}$.*

We can adjust Def. 2.3 of fully-quantum protocols to take into account the presence of errors in the training data.

**Definition A.2** (Fully-quantum protocols with noisy data). A $(\epsilon_{sp}, \epsilon_M)$-*fully-quantum protocol* for the concept class $\mathcal{C}$ in Definition 3.1 is a polynomial-time algorithm $A$ that takes as input training data of the form

$$T_{x,\epsilon_{sp},\epsilon_M} = \left\{ \left( |\psi_{f^{(i)}}\rangle_{\epsilon_{sp}}^{\otimes \ell}, (x, y, b) \right) \mid (x, y, b) \sim \pi_x^{\epsilon_M}(f^{(i)}), \right.$$
$$\left. \text{and } f^{(i)} \sim \mathcal{U}\left(\{0,1\}^N\right) \right\}_{i=1}^{\text{poly}(n)}, \tag{16}$$

and outputs a *classical description of a polynomial-time quantum algorithm* that on input $|\psi_f\rangle_{\epsilon_{sp}}^{\otimes \ell} \in \mathcal{S}_{\text{phase}}$ generates a sample from a distribution $\tilde{\pi}_x(f) \in \Delta(\{0,1\}^{2n+1})$.

Having noisy data, the natural modified learning condition for the fully-quantum protocol is the following.

**Definition A.3** ($(\epsilon, \epsilon_{sp}, \epsilon_M, \delta, p_{\text{succ}})$-fully-quantum learnable with noisy data). We say that $\mathcal{C}$ is $(\epsilon, \epsilon_{sp}, \epsilon_M, \delta, p_{\text{succ}})$-*fully-quantum learnable* if there exists a $(\epsilon_{sp}, \epsilon_M)$-fully-quantum protocol $A$ such that for every $\pi_x \in \mathcal{C}$, with probability at least $p_{\text{succ}}$ we have

$$Pr_{f \sim \mathcal{U}(\{0,1\}^N)} \left( ||\tilde{\pi}_x(f) - \pi_x(f)||_{TV} \leq 1 - \epsilon_{tot} \right) \geq \delta, \tag{17}$$

where $\epsilon_{tot} = \epsilon + \epsilon_{sp} + \epsilon_M$ and $\tilde{\pi}_x(f) \in \Delta(\{0,1\}^{2n+1})$ denotes the distribution that the polynomial-time quantum algorithm obtained from the learning algorithm $A$ generates samples from on input $|\psi_f\rangle^{\otimes \ell} \in \mathcal{S}_{\text{phase}}$.

We can now prove that the concept class $\mathcal{C}$ is fully-quantum learnable even in the presence of noisy data.

**Proposition A.4.** *Assuming a maximum error of $\epsilon_{sp}^{max}$ on the input states, the concept class in Definition 3.1 is $(0, \epsilon_{sp}^{max}, 0, 1)$-fully-quantum learnable.*

*Proof.* The proof follows directly from the one for the ideal case in Appendix A. The fully-quantum protocol performs exactly the same steps as in the ideal case, in this way the error in the prediction is bounded by the error in state preparation. Specifically, assuming for every $f \in \{0,1\}^N$ input states $|\psi\rangle_{\epsilon_{sp}}$ are such that:

$$d(|\psi_f\rangle, |\psi_f\rangle_{\epsilon_{sp}}) \leq \epsilon_{sp}^{max} \tag{18}$$

Then by the definition of trace distance we have that given any set of POVM $\{E_m\}$ (see Theorem 9.1 of (Nielsen & Chuang, 2010)) it holds that:

$$\sum_m |\text{Tr}[E_m(|\psi_f\rangle \langle\psi_f| - |\psi_f\rangle \langle\psi_f|_{\epsilon_{sp}})]| \leq d(|\psi_f\rangle, |\psi_f\rangle_{\epsilon_{sp}}) \tag{19}$$

Therefore, any probability distribution obtained from the measurement outcomes of the POVM $E_m$ on each state $|\psi_f\rangle_{\epsilon sp}$ will have a total variation distance of at most $d(|\psi_f\rangle, |\psi_f\rangle_{\epsilon sp})$ compared to the distribution induced by applying the same POVM $E_m$ on the ideal states $|\psi_f\rangle$. As the inequality (19) holds for any set of POVM $\{E_m\}$, it will particularly be true for the set of POVM $\Pi_x$ of Eq.(10) implemented by the fully-quantum protocol. This then concludes the proof as we previously showed that the fully-quantum protocol exactly reconstruct the target distribution $\pi_x(f)$ for each $f$ with zero error and we assumed $\epsilon_{sp}^{max}$ to be the maximum difference in trace distance between the noisy input states and the ideal ones. $\qquad\square$

Following the same reasoning as in the proof of Theorem 3.3 in Appendix B, it is straightforward to show that when $\epsilon_{sp}^{max} \leq 7/8$, the learning task becomes intractable for any measure-first protocol.

## B. Proof of Theorem 3.3

**Theorem 3.3.** *The concept class in Definition 3.1 is not $(\epsilon, \delta, p_{\text{succ}})$-measure-first learnable for $(1 - \epsilon) \cdot (1 - \delta) > 7/8$ and any $p_{\text{succ}} > 0$.*

*Proof of Theorem 3.3.* The main building block of our proof of Theorem 3.3 is a result in *one-way communication complexity* by Bar-Yossef, Jayram and Kerenidis (Bar-Yossef et al., 2004). They define a problem called *Hidden Matching* (HM). Here Alice is given a string $f \in \{0, 1\}^N$, while Bob is given a perfect matching $M$ on the set $[N]$, consisting of $N/2$ edges. Bob's goal is to output some $(i, j, f_i \oplus f_j)$ for some edge $(i, j) \in M$. Their main result is:

**Theorem B.1** (Classical hardness of HM (Bar-Yossef et al., 2004)). *Let $\mathcal{M}$ be any set of perfect matchings on $[N]$ that is pairwise edge-disjoint and satisfies $|\mathcal{M}| = \Omega(N)$. Let $\mu$ be the distribution over inputs to HM in which Alice's input is uniform in $\{0, 1\}^N$ and Bob's input is uniform in $\mathcal{M}$. Then, any deterministic one-way protocol for HM that errs with probability at most $1/8$ with respect to $\mu$ requires $\Omega(\sqrt{N})$ bits of communication.*

Suppose the concept class in Definition 3.1 is $(\epsilon, \delta, p_{\text{succ}})$-measure-first learnable using a measure-first protocol given by $(M, A)$ with $(1 - \epsilon) \cdot (1 - \delta) > 7/8$ and $p_{\text{succ}} > 0$. Throughout the proof, we will show that the existence of such a measure-first learning protocol contradicts the classical hardness of HM outlined in Theorem B.1. To do so, consider the HM problem with $\mathcal{M} = \{M_x \mid x \in \{0, 1\}^n\}$, where

$$M_x = \{(y, y \oplus x) \mid y \in \{0, 1\}^n\}. \tag{20}$$

and note that $|\mathcal{M}| = N$. To solve this instance of the HM problem Bob first generates training data $T_x^M$ as in Eq. 5. Note that Bob can do so because he has knowledge of the bitstring $x$. In particular, Bob can generate $f^{(i)}$ from $\{0, 1\}^N$, compute $R_{f^{(i)}}(x)$ and pick an element $(y, b)$ from it. Next, Alice applies the measure protocol $M$ to $|\psi_f\rangle$ for her input $f \in \{0, 1\}^N$ and sends $\widehat{\psi}_f = M(|\psi_f\rangle)$ to Bob. Finally, Bob applies $A$ on the data $T_x^M$ he generated and Alice's input $\widehat{\psi}_f$ to obtain a sample $(x, y, b) \sim \widetilde{\pi}_x(\widehat{\psi}_f)$. Since we assumed that $p_{\text{succ}} > 0$, we know that for any $x \in \{0, 1\}^n$ there must exist training data $\hat{T}_x^M$ and internal randomization of the learning algorithm $A$ such that the polynomial-time quantum algorithm output by the protocol satisfies Eq. 6. Throughout the remainder of this proof, we assume Bob fixes this to be the training data and internal randomization he uses for his input $x$ (note that Bob can do so because this does not depend on the input of Alice). Based on this fixed choice of training data and internal randomization we partition $\{0, 1\}^N = \mathrm{F}_{good}^x \sqcup \mathrm{F}_{bad}^x$, where $\mathrm{F}_{good}^x$ denotes the set of functions $f$ for which

$$||\widetilde{\pi}_x(f) - \pi_x(f)||_{TV} \leq 1 - \epsilon, \tag{21}$$

where $\widetilde{\pi}_x$ is the random function implemented by the quantum algorithm output by the protocol when using the training data $\hat{T}_x^M$ and internal randomization as above. Moreover, we note $|\mathrm{F}_{good}^x| \geq (1 - \delta) \cdot 2^n$ by Eq. 6. Finally, due to Eq. 6 we find that the probability that $(y, b) \in R_f(x)$ is at least

$$\Pr\big((y, b) \in R_f(x)\big) \geq (1 - \epsilon) \tag{22}$$

for all $f \in \mathrm{F}_{good}^x$. In conclusion, we find that the above described protocol is a *randomized* one-way communication protocol for HM with success probability at least $\epsilon$ for all inputs $(x, f)$ in the subset

$$\mathcal{X} := \bigcup_{x \in \{0, 1\}^n} \{x\} \times \mathrm{F}_{good}^x. \tag{23}$$

In the remainder of our proof, we let $A'(x, \widehat{\psi}_f)$ denote the protocol that Bob runs on his side (i.e., generating the training data $T_x^M$, running the algorithm $A$ on it, and drawing a sample from $\widetilde{\pi}_x(\widehat{\psi}_f)$). Also, we ensure Bob does so using only classical randomized computation by classically simulating the quantum algorithms. Next, we use Yao's principle to show that the above randomized one-way communication protocol implies the existence of a deterministic one-way communication protocol that errs with probability at most $(1 - \epsilon) \cdot (1 - \delta)$ with respect to $\mu$ (which would violate Theorem B.1 since $(1 - \epsilon) \cdot (1 - \delta) > 7/8$). Let $\mathcal{A}$ denote the family of *deterministic* protocols obtained by "hardwiring" all possible internal randomizations of the evaluation of $\widetilde{\pi}_x$ by $A'$, i.e.,

$$\mathcal{A} = \{A'_r(.,.) \mid r \in \{0,1\}^{\exp(n)}\}. \tag{24}$$

Also, let $\mathbb{X}$ be the random variable with values $(x, f)$ distributed according to the uniform distribution over $\mathcal{X}$, and let $\mathbb{A}$ be the random variable over $\mathcal{A}$ where the $r$ is uniformly random. Finally, we define the function $s : \mathcal{X} \times \mathcal{A} \to \mathbb{R}$ as

$$s((x, f), A'_r(.,.)) = \mathbb{1}\left[A'_r(x, \widehat{\psi}_f) \in R_f(x)\right]. \tag{25}$$

**Theorem B.2** (Yao's principle). *Let $\mathbb{A}$ be a random variable with values in $\mathbb{A}$ as defined in Eq. 24, and let $\mathbb{X}$ be a random variable with values in $\mathcal{X}$ as defined in Eq. 23. Then,*

$$\min_{(x,f)\in\mathcal{X}} \mathbb{E}\left[s((x, f), \mathbb{A})\right] \leq \max_{A'_r \in \mathcal{A}} \mathbb{E}\left[s(\mathbb{X}, A'_r)\right] \tag{26}$$

*where $s$ is the function defined in Eq.25.*

Observe that the quantity $\mathbb{E}\left[s(\mathbb{X}, A'_r)\right]$ is precisely the *success probability of the deterministic algorithm $A'_r \in \mathcal{A}$ with respect to the uniform distribution over $\mathcal{X}$*. Thus, Eq. 26 implies the existence of a deterministic algorithm $A'_r$ such that

$$\begin{aligned}
\mathbb{E}\left[s(\mathbb{X}, A'_r)\right] &= \Pr_{(x,f)\sim\mathcal{U}(\mathcal{X})}\left(A'_r(x, \widehat{\psi}_f) \in R_f(x)\right) \\
&\geq \min_{(x,f)\in\mathcal{X}} \mathbb{E}\left[s((x, f), \mathbb{A})\right].
\end{aligned} \tag{27}$$

Moreover, observe that the quantity $\min_{(x,f)\in\mathcal{X}} \mathbb{E}\left[s((x, f), \mathbb{A})\right]$ is precisely the *success probability of the randomized algorithm $A'$*, which we have previously shown to be at least $1 - \epsilon$. By combining this with Eq. 27 we find that

$$\Pr_{(x,f)\sim\mathcal{U}(\mathcal{X})}\left(A'_r(x, \widehat{\psi}_f) \in R_f(x)\right) \geq 1 - \epsilon. \tag{28}$$

Moreover, since $|\mathrm{F}^x_{\mathrm{good}}| \geq (1 - \delta) \cdot 2^n$ we find that

$$\Pr_{(x,f)\sim\mu}\left(A'_r(x, \widehat{\psi}_f) \in R_f(x)\right) \geq (1 - \epsilon) \cdot (1 - \delta). \tag{29}$$

Finally, since $(1 - \epsilon) \cdot (1 - \delta) > 7/8$, this violates the classical hardness of HM outlined in Theorem B.1.

$\square$

## C. Proof of Theorem 3.6

**Theorem 3.6.** *Let $\mathcal{S}_{\mathrm{pr}} = \{|\psi_{f^{(k)}}\rangle^{\otimes\ell} \mid f^{(k)}(.) = \mathsf{PRF}_n(k, .), \ k \in \mathcal{K}\}$, where $\mathsf{PRF}$ is a quantum-secure pseudorandom function with keys $\mathcal{K}$. The concept class in Definition 3.1 is not $(\epsilon, \delta, p_{\mathrm{succ}})$-measure-first learnable for $(1-\epsilon)\cdot(1-\delta)\cdot p_{\mathrm{succ}} > c$ for any constant $c > 7/8$ when the distribution over input states is uniform over $\mathcal{S}_{\mathrm{pr}}$.*

*Proof of Theorem 3.6.* Suppose the concept class in Definition 3.1 is $(\epsilon, \delta, p_{\mathrm{succ}})$-measure-first learnable with $p_{\mathrm{succ}} \cdot (1 - \epsilon) \cdot (1 - \delta) > c$ for a constant $c > 7/8$ when the distribution over input states is uniform over $\mathcal{S}_{\mathrm{pr}}$ using a measure-first protocol given by $(M, A)$. That is, for every $\pi_x \in \mathcal{C}$, with probability at least $p_{\mathrm{succ}}$ we have

$$\Pr_{k\sim\mathcal{U}(\mathcal{K}_n)}\left(||\widetilde{\pi}_x(f^{(k)}) - \pi_x(f^{(k)})||_{TV} \leq \epsilon\right) \geq 1 - \delta, \tag{30}$$

where $f^{(k)}(.) = \mathsf{PRF}(k, .)$ and $\widetilde{\pi}_x$ is the randomized quantum function obtained from $A$ on input of the form

$$T_x^M = \left\{ \left( \widehat{\psi}_{f^{(k)}}, (x, y, b) \right) \mid (x, y, b) \sim \pi_x(f^{(k)}) \right.$$

$$\left. \text{and } k \sim \mathcal{U}\left( \mathcal{K}_n \right) \right\}_{i=1}^{\text{poly}(n)}. \tag{31}$$

The main goal of the remainder of the proof is to show that the above assumptions violates the assumption that PRF is a quantum-secure pseudorandom function. To achieve this, we devise a quantum algorithm, denoted as $A^f$, which has query access to a function $f$, and which will exhibit a significant difference in the probability of outputting 1 when provided with either a truly random function $f$ or a pseudorandom function $f^{(k)}$. In essence, $A^f$ will train a measure-first protocol on phase states of pseudorandom functions and evaluate its performance on the provided function $f$, outputting 1 if it produces a correct sample $(x, y, b)$ with $(y, b) \in R_f(x)$. Assuming our measure-first protocol can successfully learn the concepts for phase states of pseudorandom functions, $A^f$ will most likely output 1 when $f$ is pseudorandom. Conversely, if $f$ is truly random, then based on arguments similar to those used in the proof of Theorem 3.3, the measure-first learning protocol is likely to be incorrect, leading $A^f$ to most of the time output 0. In particular, we consider the polynomial-time quantum algorithm $A^f$ that does the following:

1. Sample $x \sim \mathcal{U}(\{0, 1\}^n)$.

2. Generate a set of examples $T_x^M$ as in Eq. 31[4].

3. Use the learning algorithm $A$ with set of examples $T_x^M$ to obtain a quantum algorithm $A'$ for $\widetilde{\pi}_x$.

4. Using quantum query access to $f$ prepare $|\psi_f\rangle^{\otimes \ell}$.[5]

5. Apply $M$ to $|\psi_f\rangle^{\otimes \ell}$ to obtain $\widehat{\psi}_f = M\left( |\psi_f\rangle^{\otimes \ell} \right)$.

6. Apply $A'$ to $\widehat{\psi}_f$ to obtain a sample $(x, y, b)$ and output 1 if $y \in R_f(x)$, and 0 otherwise.

By the Eq. 30 and the paragraph leading up to it, we know that

$$\Pr_{k \sim \mathcal{U}(\mathcal{K}_n)} \left[ A^{f^{(k)}} = 1 \right] \geq p_{\text{succ}} \cdot (1 - \epsilon) \cdot (1 - \delta) > c. \tag{32}$$

On the other hand, from the classical lower bound for the HM problem in Theorem B.1, we know that

$$\Pr_{f \sim \mathcal{U}(\{0,1\}^N)} \left[ A^f(.) = 1 \right] \leq 7/8. \tag{33}$$

In particular, if Eq. 33 does not hold, then one can construct a one-way communication protocol for HM that succeeds with probability at least $7/8$ with respect to $\mu$ by having Bob perform steps $(2) - (3)$, having Alice perform steps $(4) - (5)$, and sending $\widehat{\psi}_f$ to Bob to perform the first part of step $(6)$ where they obtain the a sample $(x, y, b)$. In summary, we conclude that the measure-first protocol, when trained on phase states of pseudorandom functions, cannot generalize well to truly random functions based on the lower-bound established for the HM problem in Theorem B.1. Moreover, given our assumption that the concept class $\mathcal{C}$ in Definition 3.1 is $(\epsilon, \delta, p_{\text{succ}})$-measure-first learnable on phase states of pseudorandom states, it has to generalize well to other pseudorandom states. This implies a distinctive behavior of the "benchmarking algorithm" $A^f$ when provided with access to either a pseudorandom function $f^{(k)}$ or a truly random function $f$. In other words, we thus conclude that Eq. 32 and Eq. 33 are in contradiction with the assumption that PRF is a quantum-secure pseudorandom function.

$\square$

## D. Two examples of measure-first protocols

To elaborate on the definition of measure-first (MF) protocols (i.e., Definition 2.5) and their potential capabilities, we present two concrete MF protocols that, at first glance, appear capable of solving the learning task. However, in each case, we identify fundamental reasons for their failure, which highlight the limitations of MF protocols.

---

[4]Note that we can do so efficiently using a quantum algorithm since we only consider phase states of pseudo-random functions.
[5]This step is also efficient both for random and pseudorandom function since we suppose oracle access to $f$.

### D.1. Measure-first protocol based on classical shadows

One plausible MF protocol leverages the classical shadow framework (Huang et al., 2020). Here, the measurement strategy $M$ takes as input multiple copies of each state $|\psi_f\rangle$ and generates a classical representation $\widehat{\psi}_f$ by performing randomized measurements as prescribed by the classical shadow protocol (Huang et al., 2020). The learning algorithm $A$ can then use the training data $\hat{T}_x$ to determine $x$ and output a hypothesis $h_x$. This hypothesis would use the classical representation $\widehat{\psi}_f$ to predict the correct labels associated with $f$ by approximating the outcomes of the POVMs $E_j^x$.

However, for the hypothesis $h_x$ to generate samples $(y, b)$ that are close in total variation distance to the target distribution $\Lambda_x$, the precision required on each POVM $E_j^x$ must scale exponentially. As a result, the number of copies $N$ of $|\psi_f\rangle$ required by the classical shadow protocol to achieve such precision also grows exponentially. Therefore, this MF protocol fails due to fundamental limitations in efficiently compressing pseudorandom phase states into classical representations capable of achieving the required exponential precision. In this sense, our result can also be interpreted as a lower bound on the number of copies required for any shadow-based procedure to recover observables with exponential precision.

### D.2. Measure-first protocol with circuit learning

Another potential MF protocol could attempt to directly learn a polynomially-sized description of the circuit that prepares the pseudorandom states $|\psi_f\rangle$. By definition, pseudorandom states are efficiently preparable and always admit such a circuit. In this approach, the measurement scheme $M$ would take as input multiple copies of $|\psi_f\rangle$ and attempt to infer this circuit description. The learning algorithm $A$ would then use the training data $\hat{T}_x$ to determine $x$ and output a hypothesis $h_x$. The hypothesis $h_x$ could use the learned circuit to recreate the state $|\psi_f\rangle$ and implement the measurement $\Lambda_x$.

However, despite the existence of a polynomial-depth circuit that prepares $|\psi_f\rangle$, our results show that no efficient measurement strategy $M$ can extract a succinct classical description of this circuit. This failure arises from the inherent pseudorandomness of the states, which ensures that no efficient measurement strategy can compress the information contained in $|\psi_f\rangle$ into a usable classical description. Consequently, this MF protocol also fails to achieve the learning goal.

These examples highlight not only the challenges faced by MF protocols but also the fundamental separations between MF and fully quantum (FQ) protocols. They illustrate the difficulty of efficiently compressing pseudorandom phase states into classical descriptions that retain enough information for the machine learning task.

