# OpenReview forum: "Limitations of measure-first protocols in quantum machine learning"
_ICML.cc/2025/Conference — ICML 2025 poster_

### Official Review · Reviewer_8rQu · 2025-03-12

**Overall Recommendation:** 3

**Summary:**

This work is motivated by randomized measurement protocols to analyze the separation in quantum machine learning when processing quantum data using fully quantum operations versus measuring the input data and utilizing classical information. It highlights the limitations of measure-first protocols and provides examples demonstrating learning separations.

**Claims And Evidence:**

Yes.

**Essential References Not Discussed:**

I think all related works are cited.

**Experimental Designs Or Analyses:**

There is no experimental designs or analyses.

**Methods And Evaluation Criteria:**

Yes, it make sense.

**Other Comments Or Suggestions:**

No other comments.

**Other Strengths And Weaknesses:**

Strengths:
1. I believe the authors address a fundamental problem in quantum machine learning by comparing two main approaches: the measurement-first method and the fully quantum approach. While it is intuitively expected that the fully quantum approach would perform better, the authors provide a rigorous proof to support this claim.

Weakness:
1. The paper discusses the impact of noise on the task but lacks numerical validation.
2. It would be highly valuable if the authors could provide further insights into selecting between the fully quantum approach and the measurement-first approach for different applications. Currently, many methods employ classical shadow techniques combined with machine learning to infer properties such as expectation values with respect to observables, i.e.Tr[Oρ]. These methods have demonstrated promising results, achieving relatively high accuracy. It would be beneficial for the authors to comment on such cases and offer explanations on when to choose specific approaches beyond the HM discussed in this work.

**Questions For Authors:**

1. Why you do not provide numerical validations?

**Relation To Broader Scientific Literature:**

It contributes to the field of machine learning by providing guidance on using classical shadows to learn specific properties of quantum systems.

**Theoretical Claims:**

The proofs are rigorous and address a fundamental question in machine learning.

---

> ### Author Rebuttal · Authors · 2025-03-31
>
> **The paper discusses the impact of noise on the task but lacks numerical validation. Why do you not provide numerical validations?**
>
> We appreciate the reviewer’s suggestion regarding numerical simulations and experimental validation. We would like to mention that we are already collaborating with an experimental team on this front. However, there are significant challenges in conducting such experiments, even in ideal simulations. Moreover, we would like to clarify that this is a theoretical paper, where we provide rigorous guarantees on the robustness of our results to noise. While exploring the effects of various noise models on our protocols is certainly an interesting direction, we believe it falls outside the scope of this work. Our primary focus is on establishing a fundamental separation between MF and fully quantum protocols, rather than on noise analysis itself. Investigating additional aspects would introduce a different set of challenges and convey a different message, which is beyond the intended scope of this paper.
>
> Additionally, we would like to refer to our response to a similar comment from Reviewer G1ds: “It would be better to have discussions on implementation for demonstrating this separation.” In that response, we discuss the practical challenges involved in demonstrating or validating our separation, particularly the difficulties in demonstrating/validating that no single measure-first protocol can match the performance of the fully quantum protocol. Given these challenges, we believe that a comprehensive exploration of this issue would require a dedicated project, which falls beyond the scope of the current work.
>
> **It would be highly valuable if the authors could provide further insights into selecting between the fully quantum approach and the measurement-first approach for different applications. Currently, many methods employ classical shadow techniques combined with machine learning to infer properties such as expectation values with respect to observables, i.e., Tr[Oρ]. These methods have demonstrated promising results, achieving relatively high accuracy. It would be beneficial for the authors to comment on such cases and offer explanations on when to choose specific approaches beyond the Hamiltonian model (HM) discussed in this work.**
>
> We appreciate the reviewer’s insightful question. Our results provide a useful indicator of when MF protocols may be insufficient, particularly concerning the complexity of the quantum state family and the nature of the learning task. Specifically, our findings suggest that machine learning problems related to hard problems in (F)BQP/qpoly inherently require fully quantum protocols.
>
> An important result in quantum complexity theory [1] shows that any problem solvable with polynomial-sized quantum state advice can also be addressed using the ground state of a local Hamiltonian. This suggests that learning problems where the input consists of sufficiently complex local Hamiltonian ground states could exhibit a separation between MF and fully quantum protocols. While rigorously proving such separations remains a challenging open problem, this connection points to a promising research direction for understanding the limitations of MF protocols in physically relevant settings.
>
> The reviewer also raises an interesting point about classical shadow techniques, which have proven effective in estimating expectation values. While these methods can achieve high accuracy for specific tasks, they rely on structured measurement strategies and focus on extracting partial information from quantum states. In contrast, our results suggest that when the underlying learning problem fundamentally relies on the exponential storage capacity of quantum states, classical shadow techniques and other MF approaches may be insufficient.
>
> Overall, while MF methods, including classical shadows, are useful in many practical scenarios, our work identifies cases where a fully quantum approach is necessary. Investigating these cases further—particularly in the context of learning with local Hamiltonian ground states—is an exciting avenue for future research that likely requires a dedicated study.
>
> [1] Scott Aaronson and Andrew Drucker. A full characterization of quantum advice. Proceedings of the forty-second ACM Symposium on Theory of Computing.

---

### Official Review · Reviewer_G1ds · 2025-03-13

**Overall Recommendation:** 3

**Summary:**

This paper establishes a theoretical framework contrasting two quantum machine learning approaches: "fully-quantum" protocols that adaptively measure quantum data versus "measure-first" protocols restricted to fixed initial measurements. The authors prove that certain learning problems can be efficiently solved using fully-quantum methods but remain impossible for measure-first approaches, even when limited to efficiently preparable quantum states. This separation demonstrates that some learning tasks fundamentally require processing unmeasured quantum data, necessitating the exponential nature of quantum states. The work suggests ground states of complex local Hamiltonians as promising candidates for demonstrating this separation, with potential alternative constructions using computationally indistinguishable states. The work highlights the importance of fully quantum machine learning processing.

**Claims And Evidence:**

The claims and evidence are mainly from the theoretical perspective and are basically fine.

**Essential References Not Discussed:**

The reference is reasonable in this paper.

**Experimental Designs Or Analyses:**

The experimental design is not needed since this is a pure theory paper.

**Methods And Evaluation Criteria:**

The methods and evaluation criteria are reasonable for a theory paper.

**Other Comments Or Suggestions:**

1. Describe the classical analog of the machine learning problem consider in this paper and provide the literature.
2. Make task descriptions more accessible to ML researchers.
3. Elaborate on practical ML applications of the findings.
4. Include implementation strategies to demonstrate the separation.

**Other Strengths And Weaknesses:**

The strengths:
1. Rigorous proof of the case when fully quantum learners are more powerful than measure-and-learn protocols.
2. The work delivers insights for other topics such as ground states of complex Hamiltonians and computationally indistinguishable states.

The weaknesses:
1. The description of the task could be more accessible for the machine learning community.
2. It needs more discussion about the practical relevance of the main results to machine learning.
3. It would be better to have discussions on implementation for demonstrating this separation.

**Questions For Authors:**

1. Could you revise the description of your learning task to make it more accessible to researchers from the machine learning community?
2. How do your theoretical results on quantum-classical separation translate to practical applications in machine learning?
3. Are there other learning problems (more operational) that would exhibit similar performance gaps between fully-quantum and measure-first approaches?
4. What implementation strategies would you suggest for experimentally demonstrating the separation between fully-quantum and measure-first protocols?

**Relation To Broader Scientific Literature:**

Reasonable literature review.

**Theoretical Claims:**

Via a rough check of the proofs, they look correct to me.

---

> ### Author Rebuttal · Authors · 2025-03-31
>
> **The description of the task could be more accessible for the machine learning community.**
>
> We regret that the reviewer finds the description insufficiently accessible to the ML audience and take their criticism to heart. We have made a concerted effort to present a clear explanation in Section 1.1, but the interdisciplinary nature of our work and the page limits of ICML make further simplification challenging without introducing excessive quantum computing jargon. Given these constraints, we believe we have provided the most accessible presentation possible.
>
> **It needs more discussion about the practical relevance of the main results to machine learning.**
>
> While our work is primarily theoretical, we have discussed its practical relevance in the conclusion section of the paper. To summarize, our findings underscore the importance of processing unmeasured quantum data in machine learning, revealing scenarios where quantum advantages naturally arise. Specifically, we identify learning tasks that fundamentally require the exponential capacity of quantum states. While our proof relies on separations in one-way communication complexity and pseudorandom states, it suggests broader implications: quantum advice states that cannot be classically simulated with polynomial overhead -- such as the ground states of sufficiently complex local Hamiltonians -- could also demonstrate similar separations. Additionally, complexity-theoretic constructions like computationally indistinguishable states provide alternative routes to exhibiting gaps between measure-first and fully quantum protocols.
>
> **It would be better to have discussions on implementation for demonstrating this separation.**
>
> We appreciate the suggestion. We are actively collaborating with an experimental team to explore potential implementations. However, demonstrating this separation in practice poses significant challenges, even in idealized simulations. A key difficulty is determining the optimal measure-first protocol for the given task -- essential for rigorously establishing a separation. While we have ideas on tackling this, a full exploration requires a dedicated project and thus is beyond the goals we had for this work. Nevertheless, we emphasize that our proof of separation results for efficiently preparable quantum states brings our work closer to practical experimental realization.
>
> **Describe the classical analog of the machine learning problem considered in this paper and provide the relevant literature.**
>
> The problem we study is inherently quantum, making a direct classical analog difficult to define. Instead, classical analogs emerge in the learning strategies rather than the problem itself, giving rise to our distinction between measure-first and fully quantum protocols.
> In principle, the closest classical counterpart would be supervised learning with noisy or probabilistic labels, where the labeling function does not assign a deterministic label y to an input x. but instead samples y from a probability distribution p(y∣x). However, in our case, the crucial distinction is that the data x consists of quantum states. This fundamental difference -- i.e., how quantum data is processed -- underpins the separation between measure-first and fully quantum protocols, making standard results from supervised learning with noisy or probabilistic labels not directly applicable to our setting.
>
> **Are there other learning problems (more operational) that would exhibit similar performance gaps between fully quantum and measure-first approaches?**
>
> Thank you for the insightful question! Our results suggest that learning problems linked to hard problems in (F)BQP/qpoly are prime candidates for requiring fully quantum protocols. A key result by Aaronson and Drucker [1] shows that any problem solvable with polynomial-sized quantum state advice can also be solved using the ground state of a local Hamiltonian as advice. This implies that learning problems involving the ground states of sufficiently complex local Hamiltonians could also exhibit a separation between measure-first and fully quantum protocols. While rigorously proving such separations remains a challenge, studying the learning properties of these quantum states is a promising avenue for future research. We discuss this in the Conclusion section of the paper.
>
> [1] Scott Aaronson and Andrew Drucker. "A full characterization of quantum advice." Proceedings of the Forty-Second ACM Symposium on Theory of Computing.

---

### Official Review · Reviewer_StyH · 2025-03-15

**Overall Recommendation:** 1

**Summary:**

The paper compares two quantum learning paradigms: (i) the measure first protocols where the learner uses a priorly determined measurements to obtain some classical information about the training samples (shadow tomography), (ii) a fully quantum learner that is allowed to make measurements that depend on the outcomes of previously seen states. The authors show the existence of a setup where measure-first approach is provably worse than the full quantum protocols.

**Claims And Evidence:**

Yes

**Essential References Not Discussed:**

Some recent key works in shadow tomography were not cited:
- Triply efficient shadow tomography, Robbie King, David Gosset, Robin Kothari, Ryan Babbush
- Optimal tradeoffs for estimating Pauli observables, Sitan Chen, Weiyuan Gong, Qi Ye
- Adaptivity can help exponentially for shadow tomography, Sitan Chen, Weiyuan Gong, Zhihan Zhang

Especially the last paper seems to be very much related to the findings of this work.

**Experimental Designs Or Analyses:**

n/a

**Methods And Evaluation Criteria:**

N/A

**Other Comments Or Suggestions:**

no

**Other Strengths And Weaknesses:**

The results of the paper, in the context of recent works, are not surprising but useful to know. The paper is well written and rigorous.

However, I did not find the QML model confining! The training samples in the model are the quantum states with the post-measurement outcomes. This is a model rather artificial. Because of the state collapse of the quantum measurements, it is not typically feasible to have a quantum state before the measurement and the measurement outcome after that.  Unless I am missing some parts of the paper/concept. I am not sure if the presented model makes sense.

**Questions For Authors:**

Can you justify the proposed model of QML?
Isn't it more natural to consider post-measured quantum states with the measurement labels?

**Relation To Broader Scientific Literature:**

The findings are broad to the areas at the intersection of shadow tomography and quantum machine learning.

**Theoretical Claims:**

I looked at the results and it makes sense but I did not check the detailed proof arguments.

---

> ### Author Rebuttal · Authors · 2025-03-31
>
> We naturally find it disappointing that our work was evaluated with 1 out of 5. We are hopeful that the majority of this evaluation stems from misunderstandings, probably caused by our wording, which we can dispel and clarify. We will do our best to address the reviewer’s concerns and demonstrate why we believe our paper is deserving of publication at ICML.
>
> **Some recent key works in shadow tomography were not cited**
>
> Thank you for pointing out these references! We added these papers as references exhibiting the advantages that have been identified when measure-first protocols are allowed coherent measurements of multiple copies of a given quantum state (as discussed in the introduction in the manuscript). These references thus further elucidate the inherent power of measure-first protocols and further motivate the question of whether advantages are possible for fully-quantum protocols.
>
> **The results of the paper, in the context of recent works, are not surprising but useful to know.**
>
> In discussions with experts, we often encounter two opposing intuitions: some expect a distinction between fully quantum and measure-first schemes, while others—citing the success of shadow tomography—believe no such difference exists. The reviewer appears to align with the first view. To illustrate why some believe no such separation exists, we offer additional context.
>
> A significant body of evidence suggests measure-first protocols can be surprisingly powerful, making our result less obvious than it may seem. For instance, a recent milestone [2] demonstrated that a measure-first protocol (classical shadows of ground states [1]) could accurately predict many observables of gapped Hamiltonians. Similarly, a study by field experts [3] showed that for many quantum learning tasks, classical CNNs using classical shadows performed comparably to quantum neural networks. Notably, the authors questioned whether any task could conclusively separate classical and quantum approaches. The works cited by the reviewer further support the surprising strength of measure-first protocols.
>
> In the quantum ML setting, if labels corresponded to expectation values of local observables rather than measurement outcomes, measure-first and fully quantum protocols would be equivalent due to classical shadows' provable guarantees [1]. Given this, demonstrating a clear separation is nontrivial and far from obvious.
>
> Of course, the reviewer may still find the result unsurprising, but even so, we believe rigorously proving it remains a valuable contribution. Moreover, the proof is nontrivial, and we see no simpler way to establish this fact.
>
> We hope this clarifies our perspective and underscores the broader significance of our work.
>
> [1] Huang et al. "Provably Eff. ML for Quantum Many-Body Problems." Science
>
> [2] Huang, et al. "Quantum Adv. in Learning from Experiments." Science
>
> [3] Bermejo, Pablo, et al. arXiv:2408.12739.
>
> **Can you justify the proposed model of QML? Isn't it more natural to consider post-measured quantum states with the measurement labels?**
>
> We appreciate the reviewer’s concern. From their comments, we understand that “QML model” refers to supervised learning with quantum states as data and classical labels.
>
> This model is well-established in the literature as a natural generalization of classical supervised learning. The study of learning with quantum data dates back to the early 2000s [1,2], with subsequent works generalizing the framework [3,4]. More broadly, learning from quantum states is a central theme in quantum information, akin to classical supervised learning, where the goal is to learn a labeling function. Fundamental results [5] have shown exponential advantages using quantum memory, while classical shadows further motivate our approach.
>
> A key practical motivation is its relevance to experimental quantum systems. An experimenter can repeatedly prepare a quantum state ρ, measure different observables, and obtain datasets consisting of copies of ρ alongside measurement outcomes. This setup aligns with standard approaches in quantum state and shadow tomography [1,2] and formal frameworks for learning with quantum datasets [4].
>
> Finally, if the dataset were modified to include post-measurement states with their labels, it would precisely correspond to the information accessible to a measure-first protocol, which receives the same data generated by a quantum state’s measurement process, along with the classical label.
>
> Thus, we believe our model is well-motivated and widely accepted in quantum machine learning.
>
> [1] Aaronson "The learnability of quantum states."
>
> [2] Huang, et al.. "Predicting many prop. of a quantum system from very few measurements." Nature Physics
>
> [3] Gambs, S. arXiv:0809.0444
>
> [4] Guță, M., et al. "Quantum learning: asymptotically optimal classification of qubit states." New Journal of Physics
>
> [5] Chen, Sitan, et al. "Exp. separations between learning with and without quantum memory." FOCS

---

### Official Review · Reviewer_KBU4 · 2025-03-16

**Overall Recommendation:** 3

**Summary:**

This paper details the difference of two approaches in the task of quantum state or quantum distribution identification. The first approach is the measure-first approach where form a given set of quantum samples, a randomized measurement is performed and then a classical representation is constructed. The second approach, fully-quantum, takes the quantum data and from it directly constructs the classical distribution, or classical image.

**Claims And Evidence:**

The paper support the claims by proper evidence. The proofs are correct up to my understanding

**Essential References Not Discussed:**

I believe authors mentioned most relevant literature to their work

**Experimental Designs Or Analyses:**

My main concern in this work is that of concrete advancement. Considering that a) quantum computing performs unitary transforms up to the measurement, b) the HM problem has been proven and c) the definition of the learning problem in Def 2.2 and 2.5, is the novelty in this paper not an direct extension of the previous results by throwing it into a slightly different context? As such is the novelty defensible?

**Methods And Evaluation Criteria:**

The authors suport their claims on a set of previously proven results in particular on the results of the Hiddem Matching problem that states that given a quantum state it is not possible to reconstruct it in polynomial time efficiently. Based on this proof the authors claim that the measure first protocol is also not a learnable function.

**Other Comments Or Suggestions:**

I would suggest to determine that instead of looking at the proposed problem as a simple extension of the previous work on how ML can actually change this problem. Is truly quantum machine learning applied to this problem helpful? Can the problem e restated in different representation or can additional information be helpful to change the findings?

**Other Strengths And Weaknesses:**

- The paper is clear enough but I had trouble to understand it as in my opinions the definitions and flow would benefit from a clearer description.
- Definitions 2.2 and 3.1 are identical (in particular equations 2 and 7)
- The fact that the fully quantum protocol can learn the mapping $\pi_x \rightarrow \vert \psi_f\rangle\rightarrow \mathcal{U}_x\vert\psi_f\rangle\rightarrow \mathcal{M} \rightarrow y\in\mathcal{U}(\mathcal{R}_f(x))$ is under the described conditions without any difficulties on the learning mechanism (implementation not considering). However the authors did not elaborate as quote "we can introduce various levels of learning" what types of learning would be actually considered real or more of a learning tasks.

**Questions For Authors:**

How would the approach change if I have distribution of states from a generator and a copy of the same state from an ensemble?

**Relation To Broader Scientific Literature:**

The paper discusses the main papers supporting the work. Just  comment Huang 202a and Huang 202b are the two same papers. Perhaps the authors were considering the work Huan 2022, Learning quantum states from their classical shadows, Nature

**Theoretical Claims:**

Yes the proofs seems to be correct

---

> ### Author Rebuttal · Authors · 2025-03-31
>
> **My main concern in this work is that of concrete advancement. ...  As such, is the novelty defensible?**
>
> We appreciate the reviewer’s comment and would like to clarify that our contribution is both novel and nontrivial.
>
> First, our work is motivated by practical ML scenarios, focusing on quantum states efficiently preparable on quantum hardware. The HM problem, by contrast, originates in communication complexity, where lower bounds do not directly apply when states can be efficiently prepared. In such cases, the preparation circuit itself can act as a message, potentially bypassing standard lower bounds.
>
> To address this, we rigorously prove that learning remains classically intractable even for efficiently preparable states. We achieve this by leveraging pseudorandom states and imposing a time efficiency constraint on the learner—a key consideration in ML. This step is crucial: without it, the separation would lack practical significance. With it, however, we extend the separation beyond the HM setting, showing the advantage persists as long as the states are sufficiently "rich."
>
> Second, unlike the traditional HM problem, we focus on average-case correctness and robustness to noise—both critical in ML but absent in HM.
>
> In summary, our contribution is both highly relevant to ML and nontrivial (e.g., it requires invoking results on pseudorandom states and Yao’s principle for average-case hardness). We hope this clarifies the novelty of our work.
>
> Additionally, while one might expect a fully quantum protocol to be inherently more powerful than measure-first protocols -- given that, as you note in (b), quantum computing enables unitary transformations before measurement -- measure-first protocols are surprisingly powerful. This makes our results far less trivial than they may seem. For further details, we refer the reviewer to our response to Reviewer StyH regarding their comment: “The results of the paper, in the context of recent works, are not surprising but useful to know”.
>
> **Just a comment: Huang 202a and Huang 202b are the same paper.**
>
> Thank you for catching this! We will correct the references in the updated manuscript.
>
> **Definitions 2.2 and 3.1 are identical (in particular, Equations 2 and 7).**
>
> While Definitions 2.2 and 3.1 share similarities, they serve different purposes. Definition 2.2 is a general formulation of the learning problem without specifying a particular distribution, while Definition 3.1 introduces the distribution \pi_x(f) that exhibits the separation between measure-first and fully quantum protocols.
>
> **However, the authors did not elaborate on what is meant by "we can introduce various levels of learning"—what types of learning would actually be considered real or meaningful learning tasks?**
>
> By "real/more of a learning task," we refer to scenarios where the value of x (that is hidden in the dataset) cannot be directly extracted from a single datapoint but instead requires a learning algorithm that processes multiple datapoints to infer x. Specific examples of such learning tasks are detailed in Appendix A.
>
> **I would suggest exploring how quantum machine learning could fundamentally reshape this problem rather than viewing it as a straightforward extension of prior work. Is quantum machine learning genuinely useful for this problem? Could the problem be reformulated in a different representation, or could additional information change the findings?**
>
> Thank you for the thoughtful suggestion! Our primary motivation is not to explore how machine learning changes this problem but rather to highlight how machine learning itself is affected by the nature of the data it receives. Our work underscores the crucial role of processing unmeasured quantum data in learning tasks, presenting a scenario where quantum advantages naturally arise. In particular, our results suggest that certain learning problems inherently require the "exponential capacity" of quantum states, a feature distinct from classical data representations.
>
> **How would the approach change if the states were generated from a distribution rather than given as copies from an ensemble?**
>
> We must admit that we do not fully understand the reviewer’s question. In our work, the quantum states are indeed "drawn/generated" from a distribution—specifically, the uniform distribution over multiple copies of pseudorandom phase states.
>
> If your question is whether the separation still holds when receiving only a single copy at a time, the answer is yes. The fully quantum protocol can already learn the problem with just a single copy at a time, whereas providing multiple copies simultaneously can only benefit the measure-first protocol. In other words, by giving multiple copies at once, we are actually favoring the measure-first protocol, making our separation result even stronger.

---

### Decision · Program_Chairs · 2025-05-01

**Decision:**

Accept (poster)

**Comment:**

This paper establishes a theoretical framework contrasting two quantum machine learning approaches: "fully-quantum" protocols that adaptively measure quantum data versus "measure-first" protocols restricted to fixed initial measurements. The theory is solid and the conceptual message is important, although the authors could use extra efforts in motivation and describing the two settings to avoid confusion.  We hope the authors could take their rebuttals to other comments also into the revision.